# Towards More Practical Adversarial Attacks on Graph Neural Networks

**Jiaqi Ma** *†
jiaqima@umich.edu

**Shuangrui Ding** ‡†
markding@umich.edu

**Qiaozhu Mei** *‡
qmei@umich.edu

## Abstract

We study the black-box attacks on graph neural networks (GNNs) under a novel and realistic constraint: attackers have access to only a subset of nodes in the network, and they can only attack a small number of them. A node selection step is essential under this setup. We demonstrate that the structural inductive biases of GNN models can be an effective source for this type of attacks. Specifically, by exploiting the connection between the backward propagation of GNNs and random walks, we show that the common gradient-based white-box attacks can be generalized to the black-box setting via the connection between the gradient and an importance score similar to PageRank. In practice, we find attacks based on this importance score indeed increase the classification loss by a large margin, but they fail to significantly increase the mis-classification rate. Our theoretical and empirical analyses suggest that there is a discrepancy between the loss and mis-classification rate, as the latter presents a diminishing-return pattern when the number of attacked nodes increases. Therefore, we propose a greedy procedure to correct the importance score that takes into account of the diminishing-return pattern. Experimental results show that the proposed procedure can significantly increase the mis-classification rate of common GNNs on real-world data without access to model parameters nor predictions.

## 1 Introduction

Graph neural networks (GNNs) [22], the family of deep learning models on graphs, have shown promising empirical performance on various applications of machine learning to graph data, such as recommender systems [27], social network analysis [12], and drug discovery [16]. Like other deep learning models, GNNs have also been shown to be vulnerable under adversarial attacks [30], which has recently attracted increasing research interest [9]. Indeed, adversarial attacks have been an efficient tool to analyze both the theoretical properties as well as the practical accountability of graph neural networks. As graph data have more complex structures than image or text data, researchers have come up with diverse adversarial attack setups. For example, there are different tasks (node classification and graph classification), assumptions of attacker's knowledge (white-box, grey-box, and black-box), strategies (node feature modification and graph structure modification), and corresponding budget or other constraints (norm of feature changes or number of edge changes).

Despite these research efforts, there is still a considerable gap between the existing attack setups and the reality. It is unreasonable to assume that an attacker can alter the input of a large proportion of nodes, and even if there is a budget limit, it is unreasonable to assume that they can attack any node as they wish. For example, in a real-world social network, the attackers usually only have access to a few bot accounts, and they are unlikely to be among the top nodes in the network; it is difficult for

the attackers to hack and alter the properties of celebrity accounts. Moreover, an attacker usually has limited knowledge about the underling machine learning model used by the platform (e.g., they may roughly know what types of models are used but have no access to the model parameters or training labels). Motivated by the real-world scenario of attacks, in this paper we study a new type of black-box adversarial attack for node classification tasks, which is more restricted and more practical, assuming that the attacker has no access to the model parameters or predictions. Our setup differs from existing work with a novel constraint on node access, where attackers only have access to a subset of nodes in the graph, and they can only manipulate a small number of them.

The proposed black-box adversarial attack requires a two-step procedure: 1) selecting a small subset of nodes to attack under the limits of node access; 2) altering the node attributes or edges under a per-node budget. In this paper, we focus on the first step and study the node selection strategy. The key insight of the proposed strategy lies in the observation that, with no access to the GNN parameters or predictions, the strong *structural inductive biases* of the GNN models can be exploited as an effective information source of attacks. The structural inductive biases encoded by various neural architectures (e.g., the convolution kernel in convolutional neural networks) play important roles in the success of deep learning models. GNNs have even more explicit structural inductive biases due to the graph structure and their heavy weight sharing design. Theoretical analyses have shown that the understanding of structural inductive biases could lead to better designs of GNN models [25, 11]. From a new perspective, our work demonstrates that such structural inductive biases can turn into security concerns in a black-box attack, as the graph structure is usually exposed to the attackers.

Following this insight, we derive a node selection strategy with a formal analysis of the proposed black-box attack setup. By exploiting the connection between the backward propagation of GNNs and random walks, we first generalize the gradient-norm in a white-box attack into a model-independent importance score similar to the PageRank. In practice, attacking the nodes with high importance scores increases the classification loss significantly but does not generate the same effect on the mis-classification rate. Our theoretical and empirical analyses suggest that such discrepancy is due to the diminishing-return effect of the mis-classification rate. We further propose a greedy correction procedure for calculating the importance scores. Experiments on three real-world benchmark datasets and popular GNN models show that the proposed attack strategy significantly outperforms baseline methods. We summarize our main contributions as follows:

1. We propose a novel setup of black-box attacks for GNNs with a constraint of limited node access, which is by far the most restricted and practical compared to existing work.

2. We demonstrate that the structural inductive biases of GNNs can be exploited as an effective information source of black-box adversarial attacks.

3. We analyze the discrepancy between classification loss and mis-classification rate and propose a practical greedy method of adversarial attacks for node classification tasks.

4. We empirically verify the effectiveness of the proposed method on three benchmark datasets with popular GNN models.

## 2 Related Work

### 2.1 Adversarial Attack on GNNs

The study of adversarial attacks on graph neural networks has surged recently. A taxonomy of existing work has been summarized by Jin et al. [9], and we give a brief introduction here. First, there are two types of machine learning tasks on graphs that are commonly studied, node-level classification and graph-level classification. We focus on the node-level classification in this paper. Next, there are a couple of choices of the attack form. For example, the attack can happen either during model training (poisoning) or during model testing (evasion); the attacker may aim to mislead the prediction on specific nodes (targeted attack) [30] or damage the overall task performance (untargeted attack) [29]; the adversarial perturbation can be done by modifying node features, adding or deleting edges, or injecting new nodes [17]. Our work belongs to untargeted evasion attacks. For the adversarial perturbation, most existing works of untargeted attacks apply global constraints on the proportion of node features or the number of edges to be altered. Our work sets a novel local constraint on node access, which is more realistic in practice: perturbation on top (e.g., celebrity) nodes is prohibited and only a small number of nodes can be perturbed. Finally, depending on the attacker's knowledge

about the GNN model, existing work can be split into three categories: white-box attacks [23, 5, 21] have access to full information about the model, including model parameters, input data, and labels; grey-box attacks [29, 30, 17] have partial information about the model and the exact setups vary in a range; in the most challenging setting, black-box attacks [6, 2, 4] can only access the input data and sometimes the black-box predictions of the model. In this work, we consider an even more strict black-box attack setup, where model predictions are invisible to the attackers. As far as we know, the only existing works that conduct untargeted black-box attacks without access to model predictions are those by Bojchevski and Günnemann [2] and Chang et al. [4]. However both of them require the access to embeddings of nodes, which are prohibited as well in our setup.

## 2.2 Structural Inductive Bias of GNNs

While having an extremely restricted black-box setup, we demonstrate that effective adversarial attacks are still possible due to the strong and explicit structural inductive biases of GNNs.

Structural inductive biases refer to the structures encoded by various neural architectures, such as the weight sharing mechanisms in convolution kernels of convolutional neural networks, or the gating mechanisms in recurrent neural networks. Such neural architectures have been recognized as a key factor for the success of deep learning models [28], which (partially) motivate some recent developments of neural architecture search [28], Bayesian deep learning [20], Lottery Ticket Hypothesis [7], etc. The natural graph structure and the heavy weight sharing mechanism grant GNN models even more explicit structural inductive biases. Indeed, GNN models have been theoretically shown to share similar behaviours as Weisfeiler-Lehman tests [14, 24] or random walks [25]. On the positive side, such theoretical analyses have led to better GNN model designs [25, 11].

Our work instead studies the negative impact of the structural inductive biases in the context of adversarial attacks: when the graph structure is exposed to the attacker, such structural information can turn into the knowledge source for an attack. While most existing attack strategies more-or-less utilize some structural properties of GNNs, they are utilized in a *data-driven manner* which requires querying the GNN model, e.g., learning to edit the graph via a trial-and-error interaction with the GNN model [6]. We formally establish connections between the structural properties and attack strategies without any queries to the GNN model.

## 3 Principled Black-Box Attack Strategies with Limited Node Access

In this section, we derive principled strategies to attack GNNs under the novel black-box setup with limited node access. We first analyze the corresponding *white-box* attack problem in Section 3.2 and then adapt the theoretical insights from the white-box setup to the black-box setup and propose a black-box attack strategy in Section 3.3. Finally, in Section 3.4, we correct the proposed strategy by taking into account of the diminishing-return effect for the mis-classification rate.

### 3.1 Preliminary Notations

We first introduce necessary notations. We denote a graph as $G = (V, E)$, where $V = \{1, 2, \ldots, N\}$ is the set of $N$ nodes, and $E \subseteq V \times V$ is the set of edges. For a node classification problem, the nodes of the graph are collectively associated with node features $X \in \mathbb{R}^{N \times D}$ and labels $y \in \{1, 2, \ldots, K\}^N$, where $D$ is the dimensionality of the feature vectors and $K$ is the number of classes. Each node $i$'s local neighborhood including itself is denoted as $\mathcal{N}_i = \{j \in V \mid (i, j) \in E\} \cup \{i\}$, and its degree as $d_i = |\mathcal{N}_i|$. To ease the notation, for any matrix $A \in \mathbb{R}^{D_1 \times D_2}$ in this paper, we refer $A_j$ to the transpose of the $j$-th row of the matrix, i.e., $A_j \in \mathbb{R}^{D_2}$.

**GNN models.** Given the graph $G$, a GNN model is a function $f_G : \mathbb{R}^{N \times D} \to \mathbb{R}^{N \times K}$ that maps the node features $X$ to output logits of each node. We denote the output logits of all nodes as a matrix $H \in \mathbb{R}^{N \times K}$ and $H = f_G(X)$. A GNN $f_G$ is usually built by stacking a certain number ($L$) of layers, with the $l$-th layer, $1 \leq l \leq L$, taking the following form:

$$H_i^{(l)} = \sigma \left( \sum_{j \in \mathcal{N}_i} \alpha_{ij} \boldsymbol{W}_l H_j^{(l-1)} \right), \tag{1}$$

where $H^{(l)} \in \mathbb{R}^{N \times D_l}$ is the hidden representation of nodes with $D_l$ dimensions, output by the $l$-th layer; $\boldsymbol{W}_l$ is a learnable linear transformation matrix; $\sigma$ is an element-wise nonlinear activation function; and different GNNs have different normalization terms $\alpha_{ij}$. For instance, $\alpha_{ij} = 1/\sqrt{d_i d_j}$ or $\alpha_{ij} = 1/d_i$ in Graph Convolutional Networks (GCN) [10]. In addition, $H^{(0)} = X$ and $H = H^{(L)}$.

**Random walks.** A random walk [13] on $G$ is specified by the matrix of transition probabilities, $M \in \mathbb{R}^{N \times N}$, where

$$M_{ij} = \begin{cases} 1/d_i, & \text{if } (i,j) \in E \text{ or } j = i, \\ 0, & \text{otherwise.} \end{cases}$$

Each $M_{ij}$ represents the probability of transiting from $i$ to $j$ at any given step of the random walk. And powering the transition matrix by $t$ gives us the $t$-step transition matrix $M^t$.

## 3.2 White-Box Adversarial Attacks with Limited Node Access

**Problem formulation.** Given a classification loss $\mathcal{L} : \mathbb{R}^{N \times K} \times \{1, \ldots, K\}^N \to \mathbb{R}$, the problem of white-box attack with limited node access can be formulated as an optimization problem as follows:

$$\max_{S \subseteq V} \quad \mathcal{L}(H, y) \tag{2}$$
$$\text{subject to} \quad |S| \leq r, d_i \leq m, \forall i \in S$$
$$H = f(\tau(X, S)),$$

where $r, m \in \mathbb{Z}^+$ respectively specify the maximum number of nodes and the maximum degree of nodes that can be attacked. Intuitively, we treat high-degree nodes as a proxy of celebrity accounts in a social network. For simplicity, we have omitted the subscript $G$ of the learned GNN classifier $f_G$. The function $\tau : \mathbb{R}^{N \times D} \times 2^V \to \mathbb{R}^{N \times D}$ perturbs the feature matrix $X$ based on the selected node set $S$ (i.e., *attack set*). Under the white-box setup, theoretically $\tau$ can also be optimized to maximize the loss. However, as our goal is to study the node selection strategy under the black-box setup, we set $\tau$ as a pre-determined function. In particular, we define the $j$-th row of the output of $\tau$ as $\tau(X, S)_j = X_j + \mathbb{1}[j \in S]\epsilon$, where $\epsilon \in \mathbb{R}^D$ is a small constant noise vector constructed by attackers' domain knowledge about the features. In other words, the same small noise vector is added to the features of every attacked node.

We use the Carlili-Wagner loss for our analysis, a close approximation of cross-entropy loss and has been used in the analysis of adversarial attacks on image classifiers [3]:

$$\mathcal{L}(H, y) \triangleq \sum_{j=1}^{N} \mathcal{L}_j(H_j, y_j) \triangleq \sum_{j=1}^{N} \max_{k \in \{1, \ldots, K\}} H_{jk} - H_{jy_j}. \tag{3}$$

**The change of loss under perturbation.** Next we investigate how the overall loss changes when we select and perturb different nodes. We define the change of loss when perturbing the node $i$ as a function of the perturbed feature vector $x$:

$$\Delta_i(x) = \mathcal{L}(f(X'), y) - \mathcal{L}(f(X), y), \text{ where } X'_i = x \text{ and } X'_j = X_j, \forall j \neq i.$$

To concretize the analysis, we consider the GCN model with $\alpha_{ij} = \frac{1}{d_i}$ in our following derivations. Suppose $f$ is an $L$-layer GCN. With the connection between GCN and random walk [25] and Assumption 1 on the label distribution, we can show that, in expectation, the first-order Taylor approximation $\widetilde{\Delta}_i(x) \triangleq \Delta_i(X_i) + (\nabla_x \Delta_i(X_i))^T (x - X_i)$ is related to the sum of the $i$-th column of the $L$-step random walk transition matrix $M^L$. We formally summarize this finding in Proposition 1.

**Assumption 1** (Label Distribution). *Assume the distribution of the labels of all nodes follows the same constant categorical distribution, i.e.,*

$$\Pr[y_j = k] = q_k, \forall j = 1, 2, \ldots, N,$$

*where $0 < q_k < 1$ for $k = 1, 2, \ldots, K$ and $\sum_{k=1}^{K} q_k = 1$. Moreover, since the classifier $f$ has been well-trained and fixed, the prediction of $f$ should capture certain relationships among the $K$ classes. Specifically, we assume the chance for $f$ predicting any node $j$ as any class $k \in \{1, \ldots, K\}$, conditioned on the node label $y_j = l \in \{1, \ldots, K\}$, confines to a certain distribution $p(k \mid l)$, i.e.,*

$$\Pr\left[\left(\operatorname*{argmax}_{c \in \{1, \ldots, K\}} H_{jc}\right) = k \mid y_j = l\right] = p(k \mid l).$$

**Proposition 1.** *For an L-layer GCN model, if Assumption 1 and a technical assumption about the GCN[4] hold, then*

$$\delta_i \triangleq \mathbb{E}\left[\widetilde{\Delta}_i\left(x\right)|_{x=\tau(X,\{i\})_i}\right] = C \sum_{j=1}^{N} [M^L]_{ji},$$

*where $C$ is a constant independent of $i$.*

### 3.3 Adaptation from the White-Box Setup to the Black-Box Setup

Now we turn to the *black-box* setup where we have no access to the model parameters or predictions. This means we are no longer able to evaluate the objective function $\mathcal{L}(H, y)$ of the optimization problem (2). Proposition 1 shows that the relative ratio of $\delta_i/\delta_j$ between different nodes $i \neq j$ only depends on the random walk transition matrix, which we can easily calculate based on the graph $G$. This implies that we can still approximately optimize the problem (2) in the black-box setup.

**Node selection with importance scores.** Consider the change of loss under the perturbation of a set of nodes $S$. If we write the change of loss as a function of the perturbed features and take the first order Taylor expansion, which we denote as $\delta$, we have $\delta = \sum_{i \in S} \delta_i$. Therefore $\delta$ is maximized by the set of $r$ nodes with degrees less than $m$ and the largest possible $\delta_i$, where $m, r$ are the limits of node access defined in the problem (2). Therefore, we can define an *importance score* for each node $i$ as the sum of the $i$-th column of $M^L$, i.e., $I_i = \sum_{j=1}^{N} [M^L]_{ji}$, and simply select the nodes with the highest importance scores to attack. We denote this strategy as **RWCS** (Random Walk Column Sum). We note that RWCS is similar to PageRank. The difference between RWCS and PageRank is that the latter uses the stationary transition matrix $M^\infty$ for a random walk with restart.

Empirically, RWCS indeed significantly increases the classification loss (as shown in Section 4.2). The nonlinear loss actually increases linearly w.r.t. the perturbation strength (the norm of the perturbation noise $\epsilon$) for a wide range, which indicates that $\widetilde{\Delta}_i$ is a good approximation of $\Delta_i$. Surprisingly, RWCS fails to continue to increase the mis-classification rate (which matters more in real applications) when the perturbation strength becomes larger. Details of this empirical finding are shown in Figure 1 in Section 4.2. We conduct additional formal analyses on the mis-classification rate in the following section and find a diminishing-return effect of adding more nodes to the attack set when the perturbation strength is adequate.

### 3.4 Diminishing-Return of Mis-classification Rate and its Correction

**Analysis of the diminishing-return effect.** Our analysis is based on the investigation that each target node $i \in V$ will be mis-classified as we increase the attack set.

To assist the analysis, we first define the concepts of *vulnerable function* and *vulnerable set* below.

**Definition 1** (Vulnerable Function). *We define the vulnerable function $g_i : 2^V \to \{0, 1\}$ of a target node $i \in V$ as, for a given attack set $S \subseteq V$,*

$$g_i(S) = \begin{cases} 1, & \text{if } i \text{ is mis-classified when attacking } S, \\ 0, & \text{if } i \text{ is correctly-classified when attacking } S. \end{cases}$$

**Definition 2** (Vulnerable Set). *We define the vulnerable set of a target node $i \in V$ as a set of all attack sets that could lead $i$ to being mis-classified:*

$$A_i \triangleq \{S \subseteq V \mid g_i(S) = 1\}.$$

We also make the following assumption about the vulnerable function.

**Assumption 2.** *$g_i$ is non-decreasing for all $i \in V$, i.e., if $T \subseteq S \subseteq V$, then $g_i(T) \leq g_i(S)$.*

With the definitions above, the mis-classification rate can be written as the average of the vulnerable functions: $h(S) = \frac{1}{N} \sum_{i=1}^{N} g_i(S)$. By Assumption 2, $h$ is also clearly non-decreasing.

We further define the *basic vulnerable set* to characterize the minimal attack sets that can lead a target node to being mis-classified.

**Definition 3** (Basic Vulnerable Set). $\forall i \in V$, we call $B_i \subseteq A_i$ a basic vulnerable set of $i$ if,

    1) $\emptyset \notin B_i$; if $\emptyset \in A_i$, $B_i = \emptyset$;

    2) if $\emptyset \notin A_i$, for any nonempty $S \in A_i$, there exists a $T \in B_i$ s.t. $T \subseteq S$;

    3) for any distinct $S, T \in B_i$, $|S \cap T| < \min(|S|, |T|)$.

And the existence of such a basic vulnerable set is guaranteed by Lemma 1.

**Lemma 1.** *For any $i \in V$, there exists a unique $B_i$.*

The distribution of the sizes of the element sets of $B_i$ is closely related to the perturbation strength on the features. When the perturbation is small, we may have to perturb multiple nodes before the target node is mis-classified, and thus the element sets of $B_i$ will be large. When perturbation is relatively large, we may be able to turn a target node to be mis-classified by perturbing a single node, if chosen wisely. In this case $B_i$ will have a lot of singleton sets.

Our following analysis (Proposition 2) shows that $h$ has a diminishing-return effect if the vulnerable sets of nodes on the graph present *homophily* (Assumption 3), which is common in real-world networks, and the perturbation on features becomes considerably large (Assumption 4).

**Assumption 3** (Homophily). *$\forall S \in \cup_{i=1}^{N} A_i$ and $|S| > 1$, there are $b(S) \geq 1$ nodes s.t., for any node $j$ among these nodes, $S \in A_j$.*

Intuitively, the vulnerable sets present strong homophily if $b(S)$'s are large.

**Assumption 4** (Considerable Perturbation). *$\forall S \in \cup_{i=1}^{N} A_i$ and if $|S| > 1$, then there are $\lceil p(S) \cdot b(S) \rceil$ nodes s.t., for any node $j$ among these nodes, there exists a set $T \subseteq S$, $|T| = 1$, and $T \in A_j$. And $\frac{r}{r+1} < p(S) \leq 1$.*

**Proposition 2.** *If Assumptions 3 and 4 hold, $h$ is $\gamma$-approximately submodular for some $0 < \gamma < \frac{1}{r}$, i.e., there exists a non-decreasing submodular function $\tilde{h} : 2^V \to \mathbb{R}_+$, s.t. $\forall S \subseteq V$,*

$$(1 - \gamma)\tilde{h}(S) \leq h(S) \leq (1 + \gamma)\tilde{h}(S).$$

As greedy methods are guaranteed to enjoy a constant approximation ratio for such approximately submodular functions [8], Proposition 2 motivates us to develop a greedy correction procedure to compensate the diminishing-return effect when calculating the importance scores.

**The greedy correction procedure.** We propose an iterative node selection procedure and apply two greedy correction steps on top of the RWCS strategy, motivated by Assumption 3 and 4.

To accommodate Assumption 3, after each node is selected into the attack set, we exclude a $k$-hop neighborhood of the selected node for next iteration, for a given constant integer $k$. The intuition is that nodes in a local neighborhood may contribute to similar target nodes due to homophily. To accommodate Assumption 4, we adopt an adaptive version of RWCS scores. First, we binarize the $L$-step random walk transition matrix $M^L$ as $\widetilde{M}$, i.e.,

$$\left[\widetilde{M}\right]_{ij} = \begin{cases} 1, & \text{if } [M^L]_{ij} \text{ is among Top-}l \text{ of } [M^L]_i \text{ and } [M^L]_{ij} \neq 0, \\ 0, & \text{otherwise,} \end{cases} \tag{4}$$

where $l$ is a given constant integer. Next, we define a new adaptive influence score as a function of a matrix $Q$: $\widetilde{I}_i(Q) = \sum_{j=1}^{N}[Q]_{ji}$. In the iterative node selection procedure, we initialize $Q$ as $\widetilde{M}$. We select the node with highest score $\widetilde{I}_i(Q)$ subsequently. After each iteration, suppose we have selected the node $i$ in this iteration, we will update $Q$ by setting to zero for all the rows where the elements of the $i$-th column are 1. The underlying assumption of this operation is that, adding $i$ to the selected set is likely to mis-classify all the target nodes corresponding to the aforementioned rows, which

complies Assumption 4. We name this iterative procedure as the **GC-RWCS** (Greedily Corrected RWCS) strategy, and summarize it in Algorithm 1.

---

**Algorithm 1:** The GC-RWCS Strategy for Node Selection.

---

**Input:** number of nodes limit $r$; maximum degree limit $m$; neighbor hops $k$; binarized transition
      matrix $\widetilde{M}$; the adaptive influence score function $\widetilde{I}_i, \forall i \in V$.
**Output:** the set $S$ to be attacked.

1 Initialize the candidate set $P = \{i \in V \mid d_i \leq m\}$, and the score matrix $Q = \widetilde{M}$;
2 Initialize $S = \emptyset$;
3 **for** $t = 1, 2, \ldots, r$ **do**
4      $z \leftarrow \operatorname{argmax}_{i \in P} \widetilde{I}_i(Q)$;
5      $S \leftarrow S \cup \{z\}$;
6      $P \leftarrow P \setminus \{i \in P \mid \text{shortest-path}(i, z) \leq k\}$;
7      $q \leftarrow Q_{\cdot, z}$;
8      **for** $i \in V$ **do**
9          **if** $q_i$ *is* 1 **then**
10             $Q_i \leftarrow \mathbf{0}$;

11 **return** $S$;

---

Finally, we want to mention that, while the derivation of RWCS and GC-RWCS requires the knowledge of the number of layers $L$ for GCN, we find that the empirical performance of the proposed attack strategies are not sensitive w.r.t. the choice of $L$. Therefore, the proposed methods are applicable to the black-box setup where we do not know the exact $L$ of the model.

## 4 Experiments

### 4.1 Experiment Setup

**GNN models.** We evaluate the proposed attack strategies on two common GNN models, GCN [10] and JK-Net [25]. For JK-Net, we test on its two variants, JKNetConcat and JKNetMaxpool, which apply concatenation and element-wise max at last layer respectively. We set the number of layers for GCN as 2 and the number of layers for both JK-Concat and JK-Maxpool as 7. The hidden size of each layer is 32. For the training, we closely follow the hyper-parameter setup in Xu et al. [25].

**Datasets.** We adopt three citation networks, Citeseer, Cora, and Pubmed, which are standard node classification benchmark datasets [26]. Following the setup of JK-Net [25], we randomly split each dataset by 60%, 20%, and 20% for training, validation, and testing. And we draw 40 random splits.

**Baseline methods for comparison.** As we summarized in Section 2.1, our proposed black-box adversarial attack setup is by far the most restricted, and none of existing attack strategies for GNN can be applied. We compare the proposed attack strategies with baseline strategies by selecting nodes with top centrality metrics. We compare with three well-known network metrics capturing different aspects of node centrality: **Degree**, **Betweenness**, and **PageRank** and name the attack strategies correspondingly. In classical network analysis literature [15], real-world networks are shown to be fragile under attacks to high-centrality nodes. Therefore we believe these centrality metrics serve as reasonable baselines under our restricted black-box setup. For the purpose of sanity check, we also include a trivial baseline **Random**, which randomly selects the nodes to be attacked.

**Hyper-parameters for GC-RWCS.** For the proposed GC-RWCS strategy, we fix the number of step $L = 4$, the neighbor-hop parameter $k = 1$ and the parameter $l = 30$ for the binarized $\widetilde{M}$ in Eq. (4) for all models on all datasets. Note that $L = 4$ is different from the number of layers of both GCN and JK-Nets in our experiments. But we achieve effective attack performance. We also conduct a sensitivity analysis in Appendix B.2 and demonstrate the proposed method is not sensitive w.r.t. $L$.

**Nuisance parameters of the attack procedure.** For each dataset, we fix the limit on the number of nodes to attack, $r$, as 1% of the graph size. After the node selection step, we also need to specify how to perturb the node features, i.e., the design of $\epsilon$ in $\tau$ function in the optimization problem (2).

In a real-world scenario, $\epsilon$ should be designed with domain knowledge about the classification task, without access to the GNN models. In our experiments, we have to simulate the domain knowledge due to the lack of semantic meaning of each individual feature in the benchmark datasets. Formally, we construct the constant perturbation $\epsilon \in \mathbb{R}^D$ as follows, for $j = 1, 2, \ldots, D$,

$$\epsilon_j = \begin{cases} \lambda \cdot \text{sign}(\sum_{i=1}^N \frac{\partial \mathcal{L}(H,y)}{\partial X_{ij}}), & \text{if } j \in \arg \text{ top-}J\left(\left[\left|\sum_{i=1}^N \frac{\partial \mathcal{L}(H,y)}{\partial X_{il}}\right|\right]_{l=1,2,\ldots,D}\right), \\ 0, & \text{otherwise}, \end{cases} \quad (5)$$

where $\lambda$ is the magnitude of modification. We fix $J = \lfloor 0.02D \rfloor$ for all datasets. While gradients of the model are involved, we emphasize that we only use extremely limited information of the gradients: determining a few number of important features and the binary direction to perturb for each selected feature, only at the *global level* by averaging gradients on all nodes. We believe such coarse information is usually available from domain knowledge about the classification task. The perturbation magnitude for each feature is fixed as a constant $\lambda$ and is irrelevant to the model. In addition, the *same* perturbation vector is added to the features of *all* the selected nodes. The construction of the perturbation is totally independent of the selected nodes.

## 4.2 Experiment Results

**Verifying the discrepancy between the loss and the mis-classification rate.** We first provide empirical evidence for the discrepancy between classification loss (cross-entropy) and mis-classification rate. We compare the RWCS strategy to baseline strategies with varying perturbation strength as measured by $\lambda$ in Eq. (5). The results shown in Figure 1 are obtained by attacking GCN on Citeseer. First, we observe that RWCS increases the classification loss almost linearly as $\lambda$ increases, indicating our approximation of the loss by first-order Taylor expansion actually works pretty well in practice. Not surprisingly, RWCS performs very similarly as PageRank. And RWCS performs much better than other centrality metrics in increasing the classification loss, showing the effectiveness of Proposition 1. However, we see the decrease of classification accuracy when attacked by RWCS (and PageRank) quickly saturates as $\lambda$ increases. The GC-RWCS strategy that is proposed to correct the importance scores is able to decreases the classification accuracy the most as $\lambda$ becomes larger, although it increases the classification loss the least.

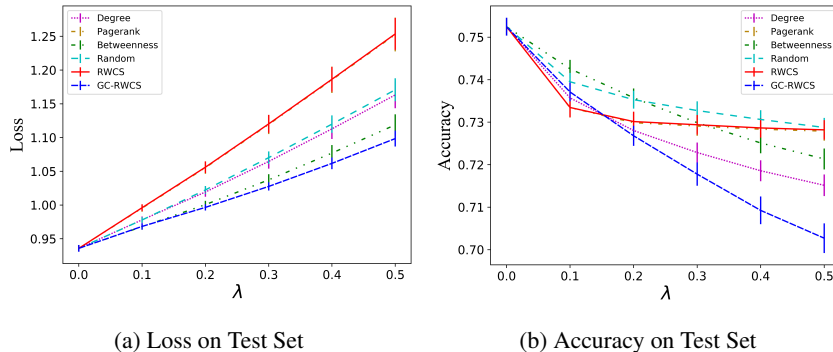

(a) Loss on Test Set        (b) Accuracy on Test Set

Figure 1: Experiments of attacking GCN on Citeseer with increasing perturbation strength $\lambda$. Results are averaged over 40 random trials and error bars indicate standard error of mean.

**Full experiment results.** We then provide the full experiment results of attacking GCN, JKNetConcat, and JKNetMaxpool on all three datasets in Table 1. The perturbation strength is set as $\lambda = 1$. The thresholds 10% and 30% indicate that we set the limit on the maximum degree $m$ as the lowest degree of the top 10% and 30% nodes respectively.

The results clearly demonstrate the effectiveness of the proposed GC-RWCS strategy. GC-RWCS achieves the best attack performance on almost all experiment settings, and the difference to the second-best strategy is significant in almost all cases. It is also worth noting that the proposed GC-RWCS strategy is able to decrease the node classification accuracy by up to 33.5%, and GC-RWCS achieves a 70% larger decrease of the accuracy than the Random baseline in most cases (see Table 3 in Appendix B.2). And this is achieved by merely adding the same constant perturbation vector to the features of 1% of the nodes in the graph. This verifies that the explicit structural inductive biases of GNN models make them vulnerable even in the extremely restricted black-box attack setup.

Table 1: Summary of the attack performance. The lower the accuracy (in %) the better the attacks. The **bold** marker denotes the best performance. The asterisk (*) means the difference between the best strategy and the second-best strategy is statistically significant by a t-test at significance level 0.05. The error bar ($\pm$) denotes the standard error of the mean by 40 independent trials.

| Method | Cora | | | Citeseer | | | Pubmed | | |
|---|---|---|---|---|---|---|---|---|---|
| | GCN | JKNetConcat | JKNetMaxpool | GCN | JKNetConcat | JKNetMaxpool | GCN | JKNetConcat | JKNetMaxpool |
| None | 85.6 ± 0.3 | 86.2 ± 0.2 | 85.8 ± 0.3 | 75.1 ± 0.2 | 72.9 ± 0.3 | 73.2 ± 0.3 | 85.7 ± 0.1 | 85.8 ± 0.1 | 85.7 ± 0.1 |
| Threshold 10% | | | | | | | | | |
| Random | 81.3 ± 0.3 | 68.8 ± 0.8 | 68.8 ± 1.3 | 71.3 ± 0.3 | 60.8 ± 0.8 | 61.7 ± 0.9 | 82.0 ± 0.3 | 75.9 ± 0.7 | 75.4 ± 0.7 |
| Degree | **78.2 ± 0.4** | 60.7 ± 1.0 | 59.9 ± 1.5 | 67.5 ± 0.4 | 52.5 ± 0.8 | 53.7 ± 1.0 | 78.9 ± 0.5 | 63.4 ± 1.0 | 63.3 ± 1.2 |
| Pagerank | 79.4 ± 0.4 | 71.6 ± 0.6 | 70.0 ± 1.0 | 70.1 ± 0.3 | 61.5 ± 0.5 | 62.6 ± 0.6 | 80.3 ± 0.3 | 71.3 ± 0.8 | 71.2 ± 0.8 |
| Betweenness | 79.7 ± 0.4 | 60.5 ± 0.9 | 60.3 ± 1.6 | 68.9 ± 0.3 | 53.5 ± 0.8 | 55.1 ± 1.0 | 78.5 ± 0.6 | 67.1 ± 1.1 | 66.2 ± 1.1 |
| RWCS | 79.5 ± 0.3 | 71.2 ± 0.5 | 69.9 ± 1.0 | 69.9 ± 0.3 | 60.8 ± 0.6 | 62.2 ± 0.7 | 79.8 ± 0.3 | 70.7 ± 0.8 | 70.7 ± 0.8 |
| GC-RWCS | 78.5 ± 0.5 | **52.7 ± 1.0\*** | **53.3 ± 1.9\*** | **65.1 ± 0.5\*** | **46.6 ± 0.8\*** | **48.2 ± 1.1\*** | **77.3 ± 0.7** | **62.1 ± 1.2** | **60.6 ± 1.4\*** |
| Threshold 30% | | | | | | | | | |
| Random | 82.6 ± 0.4 | 70.7 ± 1.1 | 71.8 ± 1.1 | 72.6 ± 0.3 | 62.7 ± 0.8 | 63.9 ± 0.8 | 82.6 ± 0.2 | 77.3 ± 0.4 | 77.4 ± 0.5 |
| Degree | **80.7 ± 0.4** | 64.9 ± 1.4 | 67.0 ± 1.5 | 70.4 ± 0.4 | 56.9 ± 0.8 | 58.7 ± 0.9 | 81.5 ± 0.4 | 72.4 ± 0.7 | 72.3 ± 0.7 |
| Pagerank | 82.6 ± 0.3 | 79.6 ± 0.4 | 79.7 ± 0.4 | 72.9 ± 0.2 | 70.2 ± 0.3 | 70.3 ± 0.3 | 83.0 ± 0.2 | 79.3 ± 0.3 | 79.6 ± 0.3 |
| Betweenness | 81.8 ± 0.4 | 64.1 ± 1.3 | 65.9 ± 1.4 | 70.7 ± 0.3 | 56.3 ± 0.8 | 58.3 ± 0.9 | 81.3 ± 0.3 | 74.1 ± 0.5 | 74.6 ± 0.5 |
| RWCS | 82.8 ± 0.3 | 79.3 ± 0.5 | 79.5 ± 0.4 | 72.9 ± 0.2 | 69.8 ± 0.3 | 70.1 ± 0.3 | 82.1 ± 0.2 | 77.8 ± 0.3 | 78.4 ± 0.3 |
| GC-RWCS | **80.7 ± 0.5** | **59.1 ± 1.6\*** | **61.1 ± 1.6\*** | **67.8 ± 0.5\*** | **49.0 ± 0.9\*** | **50.7 ± 1.1\*** | **80.3 ± 0.5\*** | **69.2 ± 0.7\*** | **70.0 ± 0.7\*** |

## 5 Conclusion

In this paper, we propose a novel black-box adversarial attack setup for GNN models with constraint of limited node access, which we believe is by far the most restricted and realistic black-box attack setup. Nonetheless, through both theoretical analyses and empirical experiments, we demonstrate that the strong and explicit structural inductive biases of GNN models make them still vulnerable to this type of adversarial attacks. We also propose a principled attack strategy, GC-RWCS, based on our theoretical analyses on the connection between the GCN model and random walk, which corrects the diminishing-return effect of the mis-classification rate. Our experimental results show that the proposed strategy significantly outperforms competing attack strategies under the same setup.

### Acknowledgments

This work was in part supported by the National Science Foundation under grant numbers 1633370 and 1620319.

## Broader Impact

For the potential positive impacts, we anticipate that the work may raise the public attention about the security and accountability issues of graph-based machine learning techniques, especially when they are applied to real-world social networks. Even without accessing any information about the model training, the graph structure alone can be exploited to damage a deep learning framework with a rather executable strategy.

On the potential negative side, as our work demonstrates that there is a chance to attack existing GNN models effectively without any knowledge but a simple graph structure, this may expose a serious alert to technology companies who maintain the platforms and operate various applications based on the graphs. However, we believe making this security concern transparent can help practitioners detect potential attack in this form and better defend the machine learning driven applications.

## Footnotes

*School of Information, University of Michigan, Ann Arbor, Michigan, USA

‡Department of EECS, University of Michigan, Ann Arbor, Michigan, USA

[4]This is an assumption made by Xu et al. [25], which we list as Assumption 5 in Appendix A.1.

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
