[Supplementary Material]

# A Proofs

## A.1 Proof of Proposition 1

We first remind the reader for some notations, a GCN model is denoted as a function $f$, the feature matrix is $X \in \mathbb{R}^{N \times D}$, and the output logits $H = f(X) \in \mathbb{R}^{N \times K}$. The $L$-step random walk transition matrix is $M^L$. More details can be found in in Section 3.1

We give in Lemma 2 the connection between GCN models and random walks. Lemma 2 relies on a technical assumption about the GCN model (Assumption 5) and the proof can be found in Xu et al. [25].

**Assumption 5** (Xu et al. [25])**.** *All paths in the computation graph of the given GCN model are independently activated with the same probability of success $\rho$.*

**Lemma 2.** *(Xu et al. [25].) Given an $L$-layer GCN with averaging as $\alpha_{i,j} = 1/d_i$ in Eq. 1, assume that all path in the computation graph of the model are activated with the same probability of success $\rho$ (Assumption 5). Then, for any node $i, j \in V$,*

$$\mathbb{E}\left[\frac{\partial H_j}{\partial X_i}\right] = \rho \cdot \prod_{l=L}^{1} W_l [M^L]_{ji}, \tag{6}$$

*where $W_l$ is the learnable parameter at $l$-th layer.*

Then we are able to prove Proposition 1 below.

*Proof.* First, we derive the gradient of the loss $\mathcal{L}(H, y)$ w.r.t. the feature $X_i$ of node $i$,

$$\nabla_{X_i} \mathcal{L}(H, y) = \nabla_{X_i} \left( \sum_{j=1}^{N} \mathcal{L}_j(H_j, y_j) \right)$$

$$= \sum_{j=1}^{N} \nabla_{X_i} \mathcal{L}_j(H_j, y_j)$$

$$= \sum_{j=1}^{N} \left( \frac{\partial H_j}{\partial X_i} \right)^T \frac{\partial \mathcal{L}_j(H_j, y_j)}{\partial H_j}, \tag{7}$$

where $H_j$ is the $j$th row of $H$ but being transposed as column vectors and $y_j$ is the true label of node $j$. Note that $\frac{\partial \mathcal{L}_j(H_j, y_j)}{\partial H_j} \in \mathbb{R}^K$, and $\frac{\partial H_j}{\partial X_i} \in \mathbb{R}^{K \times D}$.

Next, we plug Eq. 7 into $\widetilde{\Delta}_i(x)|_{x=\tau(X, \{i\})_i}$. For simplicity, We write $\widetilde{\Delta}_i(x)|_{x=\tau(X, \{i\})_i}$ as $\widetilde{\Delta}_i$ in the rest of the proof.

$$\widetilde{\Delta}_i = (\nabla_{X_i} \mathcal{L}(H, y))^T \epsilon$$

$$= \sum_{j=1}^{N} \left( \frac{\partial \mathcal{L}_j(H_j, y_j)}{\partial H_j} \right)^T \frac{\partial H_j}{\partial X_i} \epsilon. \tag{8}$$

Denote $a^j \triangleq \frac{\partial \mathcal{L}_j(H_j, y_j)}{\partial H_j} \in \mathbb{R}^K$. From the definition of loss

$$\mathcal{L}_j(H_j, y_j) = \max_{k \in \{1, \ldots, K\}} H_{jk} - H_{jy_j},$$

we have

$$a_k^j = \begin{cases} -1, & \text{if } k = y_j \text{ and } y_j \neq \text{argmax}_{c \in \{1, \ldots, K\}} H_{jc}, \\ 1, & \text{if } k \neq y_j \text{ and } k = \text{argmax}_{c \in \{1, \ldots, K\}} H_{jc}, \\ 0, & \text{otherwise}, \end{cases}$$

for $k = 1, 2, \ldots, K$. Under Assumption 1, the expectation of each element of $a^j$ is

$$\mathbb{E}[a_k^j] = -q_k(1 - p(k \mid k)) + \sum_{w=1, w \neq k}^{K} p(k \mid w) q_w, k = 1, 2, \ldots, K$$

which is a constant independent of $H_j$ and $y_j$. Therefore, we can write

$$\mathbb{E}[a^j] = c, \forall j = 1, 2, \ldots, N,$$

where $c \in \mathbb{R}^K$ is a constant vector independent of $j$.

Taking expectation of Eq. (8) and plug in the result of Lemma 2,

$$
\begin{aligned}
\mathbb{E}\left[\widetilde{\Delta}_i\right] &\approx \mathbb{E}\left[\sum_{j=1}^{N} \left(\frac{\partial \mathcal{L}_j(H_j, y_j)}{\partial H_j}\right)^T \frac{\partial H_j}{\partial X_i}\epsilon\right] \\
&= \sum_{j=1}^{N} \mathbb{E}[a^j]^T \left(\rho \prod_{l=L}^{1} W_l [M^L]_{ji}\right)\epsilon \\
&= \left(\rho c^T \prod_{l=L}^{1} W_l \epsilon\right) \sum_{j=1}^{N} [M^L]_{ji} \\
&= C \sum_{j=1}^{N} [M^L]_{ji},
\end{aligned}
$$

where $C = \rho c^T \prod_{l=L}^{1} W_l \epsilon$ is a constant scalar independent of $i$. $\qquad\square$

## A.2  Proofs for the Results in Section 3.4

**Proof of Lemma 1.**

*Proof.* If $A_i = \emptyset$, $B_i \subseteq A_i$ so $B_i = \emptyset$. The three conditions of Definition 3 are also trivially true. Below we investigate the case $A_i \neq \emptyset$.

The existence can be given by a constructive proof. We check the nonempty elements in $A_i$ one by one with any order. If this element is a super set of any other element in $A_i$, we skip it. Otherwise, we put it into $B_i$. Then we verify that the resulted $B_i$ is a basic vulnerable set for $i$. $B_i \subseteq A_i$. For condition 1), clearly, $\emptyset \notin B_i$ and if $\emptyset \in A_i$, all nonempty elements in $A_i$ are skipped so $B_i = \emptyset$. For condition 2), given $\emptyset \notin A_i$, for any nonempty $S \in A_i$, if $S \in B_i$, the condition holds. If $S \notin B_i$, by construction, there exists a nonempty strict subset $S_1 \subset S$ and $S_1 \in A_i$. If $S_1 \in B_i$, the condition holds. If $S_1 \notin B_i$, we can similarly find a nonempty strict subset $S_2 \subset S$ and $S_2 \in A_i$. Recursively, we can get a series $S \supset S_1 \supset S_2 \supset \cdots$. As $S$ is finite, we will have a set $S_k$ that no longer has strict subset so $S_k \in B_i$. Therefore the condition holds. Condition 3) means any set in $B_i$ is not a subset of another set in $B_i$. This condition holds by construction.

Now we prove the uniqueness. Suppose there are two distinct basic vulnerable sets $B_i \neq C_i$. Without loss of generality, we assume $S \in B_i$ but $S \notin C_i$. $B_i \neq \emptyset$ so $\emptyset \notin A_i$. Further $S \in A_i$, hence $C_i \neq \emptyset$. As $S \in B_i \subseteq A_i$, $S \neq \emptyset$, and $C_i$ satisfies condition 2), there will be a nonempty $T \in C_i$ s.t. $T \subset S$. If $T \in B_i$, then condition 3) is violated for $B_i$. If $T \notin B_i$, there will be a nonempty $T' \in B_i$ s.t. $T' \subset T$. But $T' \subset S$ also violates condition 3). By contradiction we prove the uniqueness. $\qquad\square$

In order to prove Proposition 2, we first would like to construct a submodular function that is close to $h$, with the help of Lemma 3 below.

**Lemma 3.** *If $\forall i \in V$, $B_i$ is either empty or only contains singleton sets, then $h$ is submodular.*

*Proof.* We first prove the case when $\forall i \in V, A_i \neq \emptyset$.

First, we show that $\forall i \in V$, if $A_i \neq \emptyset$, for any nonempty $S \subseteq V, g_i(S) = 1$ if and only if $B_i = \emptyset$ or $\exists T \in B_i, T \subseteq S$. On one hand, if $g_i(S) = 1$, then $S \in A_i$. If $\emptyset \in A_i, B_i = \emptyset$. If $\emptyset \notin A_i$, by condition 2) of the basic vulnerable set, $\exists T \in B_i, T \subseteq S$. On the other hand, if $\exists T \in B_i, T \subseteq S$, $g_i(T) = 1$, by Assumption 2, $g_i(S) \geq g_i(T)$, so $g_i(S) = 1$. If $B_i = \emptyset$, as $A_i \neq \emptyset$, if $\emptyset \notin A_i$, the condition 2) of Definition 3 will be violated. Therefore $\emptyset \in A_i$ so $g_i(\emptyset) = 1$. Still by Assumption 2, $g_i(S) \geq g_i(\emptyset)$, so $g_i(S) = 1$.

Define a function $e : V \to 2^V$ s.t. for any node $i \in V$,

$$e(i) = \{j \in V \mid \{i\} \in B_j\}.$$

Given $B_i$ is either empty or only contains singleton sets for any $i \in V$, for any nonempty $S \subseteq V$

$$h(S) = \frac{1}{N} \sum_{i=1}^N g_i(S) \tag{9}$$

$$= \frac{1}{N} |\{j \in V \mid B_j = \emptyset \text{ or } \exists T \in B_j, T \subseteq S\}|$$

$$= \frac{1}{N} |\{j \in V \mid B_j = \emptyset \text{ or } \exists \{i\} \in B_j, i \in S\}|$$

$$= \frac{1}{N} |\{j \in V \mid B_j = \emptyset \text{ or } \exists i \in S, \{i\} \in B_j\}|$$

$$= \frac{1}{N} \left( |\cup_{i \in S} e(i)| + |\{j \in V \mid B_j = \emptyset\}| \right).$$

$|\{j \in V \mid B_j = \emptyset\}|$ is a constant independent of $S$. Therefore, maximizing $h(S)$ over $S$ with $|S| \le r$ is equivalent to maximizing $|\cup_{i \in S} e(i)|$ over $S$ with $|S| \le r$, which is a maximum coverage problem. Therefore $h$ is submodular.

The case of allowing some nodes to have empty vulnerable sets can be easily proved by removing such nodes in Eq. (9) as their corresponding vulnerable functions always equal to zero. $\square$

**Proof of Proposition 2.** For simplicity, we assume $A_i \ne \emptyset$ for any $i \in V$. The proof below can be easily adapted to the general case without this assumption, by removing the nodes with empty vulnerable sets similarly as the proof for Lemma 3.

*Proof.* $\forall i \in V$, define $\widetilde{B}_i \triangleq \{S \in B_i \mid |S| = 1\}$. We can then define a new group of vulnerable sets $\widetilde{A}_i$ on $V$ for $i \in V$. Let

$$\widetilde{A}_i = \begin{cases} 2^V, & \text{if } B_i = \emptyset, \\ \emptyset, & B_i \ne \emptyset \text{ but } \widetilde{B}_i = \emptyset, \\ \{S \subseteq V \mid \exists T \in \widetilde{B}_i, T \subseteq S\}, & \text{otherwise.} \end{cases}$$

Then it is clear that $\widetilde{B}_i$ is a valid basic vulnerable set corresponding to $\widetilde{A}_i$, for $i \in V$. If we define $\tilde{g}_i : 2^V \to \{0,1\}$ as

$$\tilde{g}_i(S) = \begin{cases} 1, & \text{if } B_i = \emptyset \text{ or } \exists T \in \widetilde{B}_i, T \subseteq S, \\ 0, & \text{otherwise,} \end{cases}$$

we can easily verify that $\tilde{g}_i$ is a valid vulnerable function corresponding to $\widetilde{A}_i$, for $i \in V$. Further let $\tilde{h} : 2^V \to \mathbb{R}_+$ as

$$\tilde{h}(S) = \frac{1}{N} \sum_{i=1}^N \tilde{g}_i(S).$$

By Lemma 3, as $\forall i \in V$, $\widetilde{B}_i$ is either empty or only contains singleton sets, we know $\tilde{h}$ is submodular.

Next we investigate the difference between $h$ and $\tilde{h}$. First, for any $S \subseteq V$, if $S \notin \cup_{i=1}^N A_i$, clearly $h(S) = \tilde{h}(S) = 0$; if $|S| \le 1$, it's easy to show $h(S) = \tilde{h}(S)$. Second, for any $S \in \cup_{i=1}^N A_i$ and $|S| > 1$, by Assumption 3, there are exactly $b$ (omitting the $S$ in $b(S)$) nodes whose vulnerable set contains $S$. Without loss of generality, let us assume the indexes of $b$ nodes are $1, 2, \ldots, b$. Then, for any node $i > b$, $g_i(S) = 0, \tilde{g}_i(S) = 0$. For node $i = 1, 2, \ldots, b$, $g_i(S) = 1$, and

$$\tilde{g}_i(S) = \begin{cases} 1, & \text{if } B_i = \emptyset \text{ or } \exists T \subseteq S, |T| = 1 \text{ and } T \in \widetilde{B}_i, \\ 0, & \text{otherwise.} \end{cases}$$

By Assumption 4, there are at least $\lceil pb \rceil$ (omitting the $S$ in $p(S)$) nodes like $j$ s.t. $\tilde{g}_j(S) = 1$. Therefore, $h(S) = \frac{b}{N}$ and $\frac{\lceil pb \rceil}{N} \le \tilde{h}(S) \le \frac{b}{N}$. Hence $1 - \frac{1}{r} < 1 \le \frac{h(S)}{\tilde{h}(S)} \le \frac{1}{p} < 1 + \frac{1}{r}$. $\square$

# B Addtional Experiment Details

## B.1 Additional Dataset Details

**Datasets.** We adopt the Deep Graph Library [19] version of Cora, Citeseer, and Pubmed in our experiments. The summary statistics of the datasets are summarized in Table 2. The number of edges does not include self-loops.

Table 2: Summary statistics of datasets.

| Dataset | Nodes | Edges | Classes | Features |
|---------|-------|-------|---------|----------|
| Citeseer | 3,327 | 4,552 | 6 | 3,703 |
| Cora | 2,708 | 5,278 | 7 | 1,433 |
| Pubmed | 19,717 | 44,324 | 3 | 500 |

## B.2 Additional Experiment Results

In this section, we provide results of more experiment setups.

**Comparison to the Random baseline.** We first highlight the relative decrease of accuracy between the proposed GC-RWCS strategy ($L = 4$) and the Random strategy in Table 3. GC-RWCS is able to decrease the node classification accuracy by up to 33.5%, and achieves a 70% larger decrease of the accuracy than the Random baseline in most cases. As the GC-RWCS and Random use exactly the same feature perturbation and the node selection step of Random does not include any information of the graph structure, this relative comparison can be roughly viewed as an indicator of the attack effectiveness attributed to the structural inductive biases of the GNN models.

Table 3: Accuracy decrease (in %) comparison with clean dataset

| Method | Cora | | | Citeseer | | | Pubmed | | |
|--------|------|-------------|-------------|------|-------------|-------------|------|-------------|-------------|
| | GCN | JKNetConcat | JKNetMaxpool | GCN | JKNetConcat | JKNetMaxpool | GCN | JKNetConcat | JKNetMaxpool |
| Threshold 10% | | | | | | | | | |
| Random | 4.3 | 17.4 | 17 | 3.8 | 12.1 | 11.5 | 3.7 | 9.9 | 10.3 |
| GC-RWCS | 7.1 | 33.5 | 32.5 | 10.0 | 26.3 | 25.0 | 8.4 | 23.7 | 25.1 |
| GC-RWCS/Random | 165.12% | 192.53% | 191.18% | 263.16% | 217.36% | 217.39% | 227.03% | 239.39% | 243.69% |
| Threshold 30% | | | | | | | | | |
| Random | 3.0 | 15.5 | 14 | 2.5 | 10.2 | 9.3 | 3.1 | 8.5 | 8.3 |
| GC-RWCS | 4.9 | 27.1 | 24.7 | 7.3 | 23.9 | 22.5 | 5.4 | 16.6 | 15.7 |
| GC-RWCS/Random | 163.33% | 174.84% | 176.43% | 292.00% | 234.31% | 241.94% | 174.19% | 195.29% | 189.16% |

**More thresholds and sensitivity analysis with respect to $L$..** We provide a setup of $20\%$ threshold in addition to the $10\%$ and $30\%$ thresholds shown in Section 4.2, to give a better resolution of the results. And the results of threshold $20\%$ are consistent with other setups. We also conduct a sensitivity analysis of the hyper-parameter $L$ in GC-RWCS in Table 4, and show the results of GC-RWCS with $L = 3, 4, 5, 6, 7$. Note that GCN has 2 layers and the JK-Nets have 7 layers. The variations of GC-RWCS results with the provided range of $L$ are typically within $2\%$, indicating that the proposed GC-RWCS strategy does not rely on the exact knowledge of number of layers in the GNN models to be effective.

**Extra experiments on Graph Attetion Networks (GAT).** We further show that the proposed attack strategy GC-RWCS, while being derived from the GCN model, is also able to generalize well on GAT [18]. The results on GAT are shown in the Table 5.

**Extra experiments on a synthetic dataset.** We also run experiments on a synthetic dataset to show that, when sufficient domain knowledge regarding the node features is present, the proposed attack strategy can be made effective in a pure black-box fashion. We generate the synthetic dataset as follows. First, we generate a graph using the Barabási-Albert random graph model [1] with $N$ nodes, and denote the adjacency matrix as $A$. Then we sample $D$-dimensional node features, $X \in \mathbb{R}^{N \times D}$, from a multivariate normal distribution and take the absolute value elementwisely. Finally, we generate the node labels as follows. We first calculate $\tilde{Y} = \text{sigmoid}((A + I)XW)$, for some $W \in \mathbb{R}^D$. Considering a binary classification problem, we make the label $Y_i = 1$ if $\tilde{Y}_i > 0.5$, 0 otherwise. When attacking a GNN model trained on such a synthetic dataset, we assume the attackers

Table 4: Summary of the accuracy (in %) when $L = \{3, 4, 5, 6, 7\}$. The **bold number** and the asterisk (*) denotes the same meaning as Table 1. The underline marker denotes the values of GC-RWCS outperforms all the baseline.

| Method | Cora GCN | Cora JKNetConcat | Cora JKNetMaxpool | Citeseer GCN | Citeseer JKNetConcat | Citeseer JKNetMaxpool | Pubmed GCN | Pubmed JKNetConcat | Pubmed JKNetMaxpool |
|---|---|---|---|---|---|---|---|---|---|
| None | 85.6 ± 0.3 | 86.2 ± 0.2 | 85.8 ± 0.3 | 75.1 ± 0.2 | 72.9 ± 0.3 | 73.2 ± 0.3 | 85.7 ± 0.1 | 85.8 ± 0.1 | 85.7 ± 0.1 |
| *Threshold 10%* | | | | | | | | | |
| Random | 81.3 ± 0.3 | 68.8 ± 0.8 | 68.8 ± 1.3 | 71.3 ± 0.3 | 60.8 ± 0.8 | 61.7 ± 0.9 | 82.0 ± 0.3 | 75.9 ± 0.7 | 75.2 ± 0.7 |
| Degree | 78.2 ± 0.4 | 60.7 ± 1.0 | 59.9 ± 1.5 | 67.5 ± 0.4 | 52.5 ± 0.8 | 53.7 ± 1.0 | 78.9 ± 0.5 | 63.4 ± 1.0 | 63.2 ± 1.2 |
| Pagerank | 79.4 ± 0.4 | 71.6 ± 0.6 | 70.0 ± 1.0 | 70.1 ± 0.3 | 61.5 ± 0.5 | 62.6 ± 0.6 | 80.3 ± 0.3 | 71.3 ± 0.8 | 71.2 ± 0.8 |
| Betweenness | 79.7 ± 0.4 | 60.5 ± 0.9 | 60.3 ± 1.6 | 68.9 ± 0.3 | 53.5 ± 0.8 | 55.1 ± 1.0 | 78.5 ± 0.6 | 67.1 ± 1.1 | 66.1 ± 1.1 |
| RWCS | 79.4 ± 0.4 | 71.7 ± 0.5 | 70.3 ± 0.9 | 69.9 ± 0.3 | 62.4 ± 0.4 | 63.1 ± 0.6 | 79.8 ± 0.3 | 70.7 ± 0.8 | 70.7 ± 0.8 |
| GC-RWCS-3 | 78.6 ± 0.5 | **52.1** ± 1.1* | **53.0** ± 1.9* | **64.8** ± 0.5* | **46.4** ± 0.8* | **48.2** ± 1.0* | 78.1 ± 0.6 | 62.3 ± 1.2 | 61.6 ± 1.5 |
| GC-RWCS-4 | 78.5 ± 0.5 | 52.7 ± 1.0* | 53.3 ± 1.9* | 65.1 ± 0.5* | 46.6 ± 0.8* | 48.2 ± 1.1* | **77.3** ± 0.7 | **62.1** ± 1.2 | 60.6 ± 1.4* |
| GC-RWCS-5 | 78.9 ± 0.5 | 53.5 ± 1.1* | 54.2 ± 1.9* | 65.3 ± 0.5* | 46.6 ± 0.8* | 48.4 ± 1.0* | 78.4 ± 0.5 | 64.2 ± 1.2 | 62.5 ± 1.4 |
| GC-RWCS-6 | 78.5 ± 0.5 | 54.3 ± 1.1* | 54.9 ± 1.9* | 65.5 ± 0.5* | 47.1 ± 0.8 | 48.9 ± 1.1* | 78.0 ± 0.6 | 63.7 ± 1.1 | 62.6 ± 1.4 |
| GC-RWCS-7 | **78.1** ± 0.5 | 54.2 ± 1.1* | 54.8 ± 1.9* | 66.1 ± 0.4* | 47.5 ± 0.8 | 49.3 ± 1.1* | 78.7 ± 0.5 | 64.9 ± 1.2 | 63.3 ± 1.3 |
| *Threshold 20%* | | | | | | | | | |
| Random | 82.3 ± 0.3 | 71.7 ± 1.1 | 69.8 ± 1.1 | 72.1 ± 0.3 | 62.1 ± 0.7 | 62.6 ± 0.9 | 82.6 ± 0.2 | 77.9 ± 0.5 | 77.5 ± 0.5 |
| Degree | **79.3** ± 0.4 | 64.2 ± 1.2 | 61.6 ± 1.3 | 69.2 ± 0.4 | 56.0 ± 0.8 | 56.4 ± 1.0 | 80.6 ± 0.4 | 69.5 ± 0.8 | 69.4 ± 1.0 |
| Pagerank | 80.8 ± 0.3 | 74.5 ± 0.8 | 73.0 ± 0.8 | 72.1 ± 0.3 | 68.3 ± 0.3 | 68.2 ± 0.4 | 82.2 ± 0.2 | 77.7 ± 0.4 | 77.8 ± 0.4 |
| Betweenness | 80.7 ± 0.4 | 62.2 ± 1.4 | 60.1 ± 1.4 | 70.1 ± 0.4 | 54.8 ± 0.8 | 55.8 ± 1.1 | 80.2 ± 0.4 | 72.4 ± 0.8 | 72.0 ± 0.7 |
| RWCS | 81.4 ± 0.3 | 76.8 ± 0.6 | 76.0 ± 0.6 | 72.4 ± 0.3 | 68.9 ± 0.3 | 69.0 ± 0.4 | 81.3 ± 0.2 | 76.0 ± 0.4 | 76.5 ± 0.4 |
| GC-RWCS-3 | 79.4 ± 0.5 | **57.5** ± 1.6* | **53.1** ± 1.5* | **67.1** ± 0.4* | 48.4 ± 0.9* | 49.3 ± 1.2* | **79.0** ± 0.5* | **67.4** ± 0.9* | **66.3** ± 1.0* |
| GC-RWCS-4 | 79.4 ± 0.5 | 57.5 ± 1.7* | 53.2 ± 1.4* | 67.3 ± 0.5* | **47.9** ± 0.9* | **48.8** ± 1.3* | **79.0** ± 0.5* | 67.4 ± 1.0* | **66.3** ± 1.0* |
| GC-RWCS-5 | 79.4 ± 0.5 | 59.0 ± 1.7* | 54.5 ± 1.4* | 67.3 ± 0.4* | 48.4 ± 0.9* | 49.4 ± 1.3* | 79.2 ± 0.5* | 68.5 ± 0.9 | 68.1 ± 0.9 |
| GC-RWCS-6 | 79.5 ± 0.5 | 59.3 ± 1.7 | 54.9 ± 1.5* | 68.1 ± 0.4* | 49.2 ± 0.9* | 50.2 ± 1.3* | 79.1 ± 0.5* | 68.4 ± 0.9 | 68.5 ± 1.0 |
| GC-RWCS-7 | 79.4 ± 0.5 | 59.3 ± 1.6 | 55.3 ± 1.5* | 68.1 ± 0.4* | 50.0 ± 0.9* | 50.8 ± 1.3* | 79.2 ± 0.5* | 68.7 ± 0.9 | 68.2 ± 0.8 |
| *Threshold 30%* | | | | | | | | | |
| Random | 82.6 ± 0.4 | 70.7 ± 1.1 | 71.8 ± 1.1 | 72.6 ± 0.3 | 62.7 ± 0.8 | 63.9 ± 0.8 | 82.6 ± 0.2 | 77.3 ± 0.4 | 77.3 ± 0.5 |
| Degree | 80.7 ± 0.4 | 64.9 ± 1.4 | 67.0 ± 1.5 | 70.4 ± 0.4 | 56.9 ± 0.8 | 58.7 ± 0.9 | 81.5 ± 0.4 | 72.4 ± 0.7 | 72.1 ± 0.8 |
| Pagerank | 82.6 ± 0.3 | 79.6 ± 0.4 | 79.7 ± 0.4 | 72.9 ± 0.2 | 70.2 ± 0.3 | 70.3 ± 0.3 | 83.0 ± 0.2 | 79.3 ± 0.3 | 79.5 ± 0.3 |
| Betweenness | 81.8 ± 0.4 | 64.1 ± 1.3 | 65.9 ± 1.4 | 70.7 ± 0.3 | 56.3 ± 0.8 | 58.3 ± 0.9 | 81.3 ± 0.3 | 74.1 ± 0.5 | 74.5 ± 0.5 |
| RWCS | 82.9 ± 0.3 | 79.7 ± 0.4 | 80.0 ± 0.4 | 72.9 ± 0.2 | 70.2 ± 0.3 | 70.4 ± 0.3 | 82.1 ± 0.2 | 77.8 ± 0.3 | 78.4 ± 0.3 |
| GC-RWCS-3 | **80.2** ± 0.6 | **57.3** ± 1.7* | **59.0** ± 1.6* | 67.9 ± 0.5* | 49.1 ± 0.9* | 49.8 ± 1.1* | 80.3 ± 0.5* | **69.0** ± 0.7* | **69.8** ± 0.7* |
| GC-RWCS-4 | 80.7 ± 0.5 | 59.1 ± 1.6* | 61.1 ± 1.6* | **67.8** ± 0.5* | **49.0** ± 0.9* | 50.7 ± 1.1* | 80.3 ± 0.5* | 69.2 ± 0.7* | 70.0 ± 0.7* |
| GC-RWCS-5 | 80.8 ± 0.5 | 59.8 ± 1.6* | 61.5 ± 1.6* | 68.4 ± 0.5* | 49.2 ± 0.9* | 51.2 ± 1.1* | **80.2** ± 0.5* | 70.4 ± 0.6* | 71.5 ± 0.6 |
| GC-RWCS-6 | 80.7 ± 0.5 | 59.8 ± 1.5* | 61.4 ± 1.5* | 68.5 ± 0.5* | 50.5 ± 0.9* | 52.2 ± 1.1* | **80.2** ± 0.5* | 70.5 ± 0.5* | 71.6 ± 0.6 |
| GC-RWCS-7 | 80.7 ± 0.5 | 60.2 ± 1.5* | 61.9 ± 1.5* | 68.7 ± 0.5* | 50.7 ± 0.9* | 52.6 ± 1.1* | 80.3 ± 0.4* | 70.9 ± 0.5* | 71.9 ± 0.6 |

Table 5: Accuracy (in %) on GAT model

| Dataset threshold | Cora 10% | Cora 20% | Cora 30% | Citeseer 10% | Citeseer 20% | Citeseer 30% | Pubmed 10% | Pubmed 20% | Pubmed 30% |
|---|---|---|---|---|---|---|---|---|---|
| None | | 87.8 ± 0.2 | | | 76.9 ± 0.3 | | | 85.2 ± 0.1 | |
| Random | 72.9 ± 0.5 | 73.8 ± 0.6 | 73.9 ± 0.6 | 70.0 ± 0.5 | 71.2 ± 0.4 | 71.7 ± 0.4 | 73.9 ± 0.4 | 75.4 ± 0.3 | 76.2 ± 0.3 |
| Degree | 66.5 ± 0.7 | 67.3 ± 0.7 | 69.8 ± 0.7 | 63.3 ± 0.5 | 65.9 ± 0.4 | 67.9 ± 0.3 | 66.7 ± 0.7 | 69.0 ± 0.5 | 71.2 ± 0.4 |
| Pagerank | 74.3 ± 0.5 | 74.8 ± 0.3 | 82.4 ± 0.2 | 69.5 ± 0.3 | 72.9 ± 0.3 | 74.2 ± 0.3 | 71.6 ± 0.4 | 78.1 ± 0.2 | 79.1 ± 0.2 |
| Betweenness | 64.8 ± 0.5 | 66.0 ± 0.5 | 67.3 ± 0.6 | 65.2 ± 0.5 | 66.5 ± 0.4 | 67.6 ± 0.3 | 63.4 ± 0.7 | 68.4 ± 0.6 | 72.0 ± 0.4 |
| RWCS | 71.1 ± 0.5 | 74.6 ± 0.3 | 82.5 ± 0.2 | 69.2 ± 0.3 | 72.9 ± 0.3 | 73.9 ± 0.3 | 69.4 ± 0.5 | 74.9 ± 0.3 | 77.9 ± 0.2 |
| GC-RWCS | **58.1** ± 0.6* | **57.9** ± 0.6* | **63.0** ± 0.5* | **58.3** ± 0.6* | **61.9** ± 0.6* | **61.9** ± 0.4* | **58.9** ± 0.9* | **63.8** ± 0.7* | **68.9** ± 0.5* |

know a couple of "important features" with large weights in $W$ but do not know any of the trained model information. We randomly generate two datasets with $N = 3000$ and $D = 10$. And the experiment results are shown in the Table 6. And we find the proposed GC-RWCS model performs well on these two synthetic datasets.

Table 6: Accuracy (in %) on GCN model with synthetic data

| Dataset Threshold | synthetic_0 10% | synthetic_0 20% | synthetic_0 30% | synthetic_1 10% | synthetic_1 20% | synthetic_1 30% |
|---|---|---|---|---|---|---|
| None | | 83.6±0.3 | | | 85.2±0.4 | |
| Degree | 77.9±0.4 | **77.5**±0.4 | 79.2±0.4 | **77.4**±0.4 | 79.7±0.4 | 79.9±0.2 |
| Pagerank | 76.7±0.4 | 78.3±0.3 | 79.2±0.4 | 78.5±0.5 | 79.2±0.3 | 80.1±0.3 |
| Between | 76.3±0.4 | 80.4±0.3 | 79.1±0.4 | 78.7±0.3 | 80.6±0.3 | 80.3±0.3 |
| Random | 79.3±0.4 | 80.3±0.4 | 81.2±0.4 | 82.4±0.4 | 79.6±0.3 | 82.2±0.2 |
| RWCS | 74.9±0.5 | 78.8±0.4 | 78.5±0.3 | 79.0±0.4 | 79.9±0.4 | 80.0±0.3 |
| GC-RWCS | **74.0**±0.3* | 77.9±0.5 | **77.4**±0.4* | 77.5±0.4 | **78.9**±0.3* | **78.4**±0.3* |

**Diminishing-return effect with respect to $J$.** In Section 4.2, we have empirically shown the diminishing-return effect on the mis-classification rate when strengthening the adversarial perturbation by increasing the perturbation strength $\lambda$. Here we further validate our observation by showing that a similar effect appears as we strengthen the adversarial perturbation by increasing the number of features to be perturbed, $J$. The results are shown in Figure 2.

(a) Loss on Test Set

(b) Accuracy on Test Set

Figure 2: Experiments of attacking GCN on Citeseer with increasing number of features to be perturbed, $J$. Results are averaged over 40 random trials and error bars indicate standard error of mean.