[Reviews · NeurIPS 2020]

Review 1

Summary and Contributions: This work studies a restricted black-box attack on GNN: only nodes out of the league of highest degrees can be modified, and they are modified by a constant vector. The attack assumes little knowledge of the node classification model and mostly relies on the graph structure. This is a reasonable and novel setting. Based on a random-walk analysis and an assumption of the label distribution, the authors propose a simple strategy RWCS that attacks nodes with the highest importance scores, defined as the column sum of a certain power of the transition matrix. This strategy fails to continue to increase the mis-classification rate when the perturbation strength becomes larger, and hence the authors further propose a greedy-correction RWCS, based on an analysis revealing that the mean vulnerable function is approximately submodular under a homophily assumption. The GC-RWCS strategy is shown to outperform one random and a few centrality baselines on three benchmark data sets and two GNNs.

Strengths: - The attack setting under study is carefully explained. That one may not attack high-degree nodes (celebrity nodes) appears sensible. - The proposed methods are simple to execute. - The proposed methods are motivated and backed by theoretical analysis. - This work is a valuable contribution to the GNN attack literature.

Weaknesses: - The execution of the method is not truly black-box; it requires the gradient information to compute the perturbation vector. - That the perturbation vector is constant appears to be limited. It is possible to develop a more severe attack when the perturbations are more finely curated.

Correctness: All sound correct.

Clarity: This paper is very well written.

Relation to Prior Work: The authors explain well the novelty of the work in context.

Reproducibility: Yes

Additional Feedback: - In Eqn (1), W should be on the right of H. - The maximum node degree m does not seem to appear in the analysis. Could the authors elaborate how m plays a role in theory? ----------- Post rebuttal: I have read the response and thank the authors for their efforts.


Review 2

Summary and Contributions: The paper studies a new type of black box adversarial attack for the node classification task, assuming that the attacker has no access to the model parameters and only on the input. Furthermore, a constraint is included to select the subset of nodes to attack based on their importance which is deduced by a combination of random walk and structural information on the network and graph. Alteration at the node attributes level is performed. The experimental evaluation investigates the effect of these attacks at the loss and misclassification rate level, as the perturbation strength varies

Strengths: - The application and relevance are potentially impactful, ultimately helping to understand the robustness of GNN models. - The method is well described, with a clear notation while the claims are theoretically consolidated

Weaknesses: - The experimental setup is rather limited with respect to the variety of GNN approaches tested. The paper only looks at the GCN model. Given that the effectiveness of the proposed method mainly relies on the experimental results, I'd recommend to provide a more extensive comparison. - In Figure 1 the results for PageRank are missing - The parameter k determining the neighbour is a key parameter to the selection strategy. Possibly, the authors could investigate the effect of varying it in more detail.

Correctness: The theoretical and empirical analysis are both correct. Nevertheless, I would extend the empirical comparison as it is crucial to establish the relevance and contribution of the study.

Clarity: I find the paper very clear and easy to read

Relation to Prior Work: RWCS seem to be quite similar to PageRank, maybe the specific contribution of the proposed approach could be better highlighted

Reproducibility: Yes

Additional Feedback: Thank you to the authors for their effort in the rebuttal. I updated my recommendation.


Review 3

Summary and Contributions: This work presents an adversarial attack on Graph Neural Networks. The attack is designed for scenarios limited both in the number of accessible nodes and by the black-box nature of the attack. Despite these restrictions, the work presents indications that the attack is competitive and outperforms other attack strategies in similar settings.

Strengths: The paper is well organized and concise. The assumed threat model incorporates a more realistic perspective than previous works. It presents both theoretical arguments and experimental verification in support of its major claims.

Weaknesses: - The theoretical foundation builds off a number of assumptions, specifically assumption 5, that may limit its applicability in real world scenarios. - The work claims the attack is limited by the adversaries limited access to the attack set (the vulnerable nodes of the graph). But there is no discussion on how this attack set is generated, its properties, or its effect on the proposed methodology. - The three baseline attacks used in the experimental results are not sufficiently introduced. - Further, it is not clear why the chosen baselines are more sufficient comparisons than recent works like [1] and [3]. - While the results indicate that in many cases the methodology decreases the performance of the model, in many of the scenarios there is only a minor improvement over the baseline comparisons. - Lambda, the magnitude of perturbation on each node, is used as a measure of the strength of the attack, but intuitively it seems like J, the number of node perturbed, would be a more accurate judge of the attack strength.

Correctness: Yes.

Clarity: Yes

Relation to Prior Work: Yes.

Reproducibility: Yes

Additional Feedback: The effectiveness of the attacks seems to be weaker when applied to GCN over the other two models. Do the authors have any intuitive understanding for why this occurs?


Review 4

Summary and Contributions: This paper considers the problem of black-box attacks on graph neural networks under a realistic constraint that attackers have access to only a subset of nodes.

Strengths: The paper is theoretically grounded with good empirical results.

Weaknesses: Readability of the paper can be improved.

Correctness: yes

Clarity: yes

Relation to Prior Work: yes

Reproducibility: Yes

Additional Feedback: Thank you for your effort in preparing this rebuttal. I will keep my score and recommend acceptance. +++++++++++++++++++++++++++++++++++++++++++++++++++++++++++++ The authors consider a novel setup of black-box attacks for GNNs with a constraint of limited node access. The approach seems sounds and empirical results are reasonable. It seems to me that implication of this work is to highlight the extent of vulnerability of GNNs. I could not see how the proposed approach for example can be used to design better defense schemes (e.g., adversarial training). Being said that, I feel highlighting this vulnerability is still important. I would suggest authors to also look into scenarios where feature perturbation matrix \tau is optimized (using existing schemes). Further, adding a discussion on the implication of their findings and approach on adversarial defense would be useful.

[Author Response · NeurIPS 2020]

– For **Reviewer # 1**: Thank you for the comments! We address your specific concerns in detail below. –

*Q1. Fully black-box.* We first note that the mild use of model information to identify features to perturb can be replaced
with domain knowledge in real applications (line 287-294), which will make the black-box setup strict. To further
verify this, we construct a synthetic dataset where we know which features are important for the labels. GC-RWCS
works better than all other baselines on this synthetic data without any access to the model information. The full results
cannot be shown here due to the page limit, but we will include them in the Appendix of the final draft.

*Q2. The constant perturbation vector.* The perturbation vector indeed could be optimized given more domain knowledge.
We use a constant vector to reflect "very limited attacker knowledge" and leave optimizing it as future work.

*Q3. The maximum degree.* The maximum degree $m$ does not influence the single node effect (proposition 1) but surely
has an impact on the overall attack performance. This impact is hard to quantify theoretically, which is an interesting
future direction. If one were able to quantify it, the effect of $m$ should be reflected in the greedy correction steps.

– For **Reviewer # 2**: Thank you for the comments! We address your specific concerns in detail below. –

*Q1. More GNNs.* Per your advice, we conduct the same experiments on the Graph Attention Network (GAT). We
observe a similar trend: the proposed GC-RWCS strategy is able to significantly degrade the model performance; and it
outperforms all the baseline methods (see the table below). We will incorporate these experiments in our final draft.

| Dataset: threshold | Cora: 10% | 20% | 30% | Citeseer: 10% | 20% | 30% | Pubmed: 10% | 20% | 30% |
|---|---|---|---|---|---|---|---|---|---|
| No-Attack | | $87.8 \pm 0.2$ | | | $76.9 \pm 0.3$ | | | $85.2 \pm 0.1$ | |
| Random | $72.9 \pm 0.5$ | $73.8 \pm 0.6$ | $73.9 \pm 0.6$ | $70.0 \pm 0.5$ | $71.2 \pm 0.4$ | $71.7 \pm 0.4$ | $73.9 \pm 0.4$ | $75.4 \pm 0.3$ | $76.2 \pm 0.3$ |
| Degree | $66.5 \pm 0.7$ | $67.3 \pm 0.7$ | $69.8 \pm 0.7$ | $63.3 \pm 0.5$ | $65.9 \pm 0.4$ | $67.9 \pm 0.3$ | $66.7 \pm 0.7$ | $69.0 \pm 0.5$ | $71.2 \pm 0.4$ |
| Pagerank | $74.3 \pm 0.5$ | $74.8 \pm 0.3$ | $82.4 \pm 0.2$ | $69.5 \pm 0.3$ | $72.9 \pm 0.3$ | $74.2 \pm 0.3$ | $71.6 \pm 0.4$ | $78.1 \pm 0.2$ | $79.1 \pm 0.2$ |
| Betweenness | $64.8 \pm 0.5$ | $66.0 \pm 0.5$ | $67.3 \pm 0.6$ | $65.2 \pm 0.5$ | $66.5 \pm 0.4$ | $67.6 \pm 0.3$ | $63.4 \pm 0.7$ | $68.4 \pm 0.6$ | $72.0 \pm 0.4$ |
| RWCS | $71.1 \pm 0.5$ | $74.6 \pm 0.3$ | $82.5 \pm 0.2$ | $69.2 \pm 0.3$ | $72.9 \pm 0.3$ | $73.9 \pm 0.3$ | $69.4 \pm 0.5$ | $74.9 \pm 0.3$ | $77.9 \pm 0.2$ |
| GC-RWCS | $\mathbf{58.1} \pm 0.6^*$ | $\mathbf{57.9} \pm 0.6^*$ | $\mathbf{63.0} \pm 0.5^*$ | $\mathbf{58.3} \pm 0.6^*$ | $\mathbf{61.9} \pm 0.6^*$ | $\mathbf{61.9} \pm 0.4^*$ | $\mathbf{58.9} \pm 0.9^*$ | $\mathbf{63.8} \pm 0.7^*$ | $\mathbf{68.9} \pm 0.5^*$ |

*Q2. Figure 1, and RWCS vs PageRank.* First, we clarify that PageRank is not missing but almost overlaps with RWCS
in Figure 1. RWCS is indeed very similar to PageRank (line 188). However, we also highlighted our contribution
(line 57-59) for **revealing the novel connection** between the black-box adversarial attack on GNN and the PageRank-
like heuristic, RWCS. Thanks to this connection, we further developed the practically effective strategy, GC-RWCS.

*Q3. Sensitivity of parameter $k$.* We first note that $k$ **was fixed as 1 in all experiment setups** in our paper
(see line 276). Responsively, we further conduct a sensitivity analysis of $k$. We observe similar trends on all
datasets and thresholds, and below we show results on Cora with threshold 30% due to the page limit. The re-
sults under GC-RWCS attacks with $k = 1$ and $k = 2$ are very similar. The results of the null choice $k = 0$,
i.e., not removing neighbors, are slightly worse as expected; but they are still better than all other baselines.

| Model | GCN | | | GAT | | | JKNetConcat | | |
|---|---|---|---|---|---|---|---|---|---|
| $k$ | $k=0$ | $k=1$ | $k=2$ | $k=0$ | $k=1$ | $k=2$ | $k=0$ | $k=1$ | $k=2$ |
| GC-RWCS | $81.0 \pm 0.5$ | $\mathbf{80.7} \pm 0.5$ | $\mathbf{80.7} \pm 0.5$ | $66.1 \pm 0.7$ | $63.0 \pm 0.5$ | $\mathbf{62.6} \pm 0.8$ | $65.2 \pm 1.0$ | $\mathbf{59.1} \pm 1.6$ | $62.1 \pm 1.3$ |

– For **Reviewer # 3**: Thank you for the comments! We address your specific concerns in detail below. –

*Q1. Assumption 5 is strong.* We agree that assumption 5 (which comes from [22]) seems a bit strong. However, we
believe assumption 5 approximately holds **at a coarse level**, which is enough to develop a **black-box** attack strategy.
**And our empirical results indeed seem to support our conjecture.**

*Q2. Influence of attack set constraints.* We would like to clarify that the constraints on the attack set are illustrated by
the optimization problem (2) after line 145, which are $|S| \leq r, d_i \leq m, \forall i \in S$. **The influence of these constraints**
**on the proposed method is fundamental**: these constraints define the optimization problem (2) and its black-box
counterpart (a novel black-box attack setup), which then lead to the derivation of the proposed method.

*Q3. Description of baselines.* The three baselines are strategies that "select nodes with top centrality" (line 286), with
the centrality metrics being Degree, Betweeness, and PageRank. We will add more details in final draft.

*Q4. Comparison to [1, 3].* [1] and [3] require extra model information thus not applicable in our setup (line 266-268).

*Q5. Minor improvements over baselines.* We respectfully disagree. GC-RWCS performs the best **in all but one setup**.
The difference between GC-RWCS and the best baseline is **significant in 15/18 setups**, with **up to 13%** improvement.

*Q6. $J$ as attack strength.* We verified that replacing $\lambda$ with $J$ as the measure for attack strength yields almost the same
plots in Figure 1. We are not able to include the new plots here due to the page limit but will do so in Appendix.

*Q7. Why attack is less effective on GCN than on JKNets.* Our intuition is that GCN is shallower than JKNets so less
model information is leaked by the graph structure. This phenomenon also supports our claim in line 68-69.

– For **Reviewer # 4**: Thank you for your suggestion! Adversarial defense under our novel black-box setup is indeed an
interesting future direction. For optimizing perturbation matrix please see our response to Q2 of Reviewer 1.

[Meta-Review · NeurIPS 2020]

The paper proposes a restricted black-box attach for GNN, which is claimed to be more applicable in real world scenarios. After extensive discussion and having read the reviews and the rebuttal it is clear that the novelty of the approach is acknowledged across the board. This leaves the main weakness of the work in the experimental validation. Although a fair comparison with other approaches is hard to make due to the more challenging setting of this work, an in depth analysis of the method and the various settings would have been desirable. Some of the choices seem not very realistic as well, i.e. assuming to be able to perturb 1% of the nodes.